# Non-greedy Gradient-based Hyperparameter Optimization Over Long Horizons

## Abstract

Gradient-based meta-learning has earned a widespread popularity in few-shot deep learning, but remains broadly impractical for tasks with long horizons (many gradient steps), due to memory scaling and gradient degradation issues. A common workaround is to learn meta-parameters online, but this introduces greediness which comes with a significant performance drop. In this work, we enable non-greedy meta-learning of hyperparameters over long horizons by sharing hyperparameters that are contiguous in time, and using the sign of hypergradients rather than their magnitude to indicate convergence. We implement this with forward-mode differentiation, which we extend to the popular momentum-based SGD optimizer. We demonstrate that the hyperparameters of this optimizer can be learned non-greedily without gradient degradation over $\sim 10^4$ inner gradient steps, by only requiring $\sim 10$ outer gradient steps. On CIFAR-10, we outperform greedy and random search methods for the same computational budget by nearly 10%. Code will be available upon publication.

## 1 Introduction

Deep neural networks have shown tremendous success on a wide range of applications, including classification (He et al., 2016), generative models (Brock et al., 2019), natural language processing (Devlin et al., 2018) and speech recognition (Oord et al., 2016). This success is in part due to effective optimizers such as SGD with momentum or Adam (Kingma & Ba, 2015), which require carefully tuned hyperparameters for each application. In recent years, a long list of heuristics to tune such hyperparameters has been compiled by the deep learning community, including things like: how to best decay the learning rate (Loshchilov & Hutter, 2017), how to scale hyperparameters with the budget available (Li et al., 2020), and how to scale learning rate with batch size (Goyal et al., 2017). Unfortunately these heuristics are often dataset specific and architecture dependent (Dong et al., 2020), and must constantly evolve to accommodate new optimizers (Loshchilov & Hutter, 2019), or new tools, like batch normalization which allows for larger learning rates and smaller weight decay (Ioffe & Szegedy, 2015).

With so many ways to choose hyperparameters, the deep learning community is at risk of adopting models based on how much effort went into tuning them, rather than their methodological insight. The field of hyperparameter optimization (HPO) aims to find hyperparameters that provide a good generalization performance automatically. Unfortunately, existing tools are rather unpopular for deep networks, largely owing to their inefficiency. Here we focus on gradient-based HPO, which calculates hypergradients, i.e. the gradient of some generalization loss with respect to each hyperparameter. Gradient-based HPO should be more efficient than black box methods as the dimensionality of the hyperparameter space increases, since it relies on gradients rather than trial and error. In practice however, learning hyperparameters with gradients has only been popular in few-shot learning tasks where the horizon is short. This is because long horizons cause hypergradient degradation, and incur a memory cost that makes reverse-mode differentiation prohibitive. Greedy alternatives alleviate both of these issues, but come at the cost of solving hyperparameters locally instead of globally. Forward-mode differentiation has been shown to offer a memory cost constant with horizon size, but it doesn't address gradient degradation and only scales to few hyperparameters, which has limited its use to the greedy setting as well.

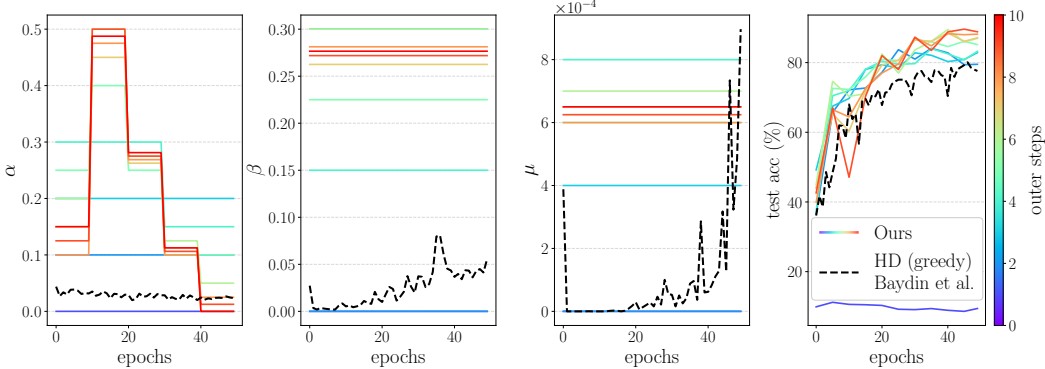

Figure 1: Our method applied to the SGD optimizer to learn (from left to right) the learning rate schedule $\alpha$, the momentum $\beta$, and weight decay $\mu$ for a WRN-16-1 on CIFAR-10. For each outer step (color) we solve CIFAR-10 from scratch for 50 epochs, and update all hyperparameters such that the final weights minimize some validation loss. We use hyperparameter sharing over 10, 50 and 50 epochs for $\alpha$, $\beta$ and $\mu$ respectively. All hyperparameters are initialized to zero and converge within just 10 outer steps to values that significantly outperform the online greedy alternative (Baydin et al., 2018), and match aggressively hand-tuned baselines for this setting (see Section 5.2).

To the best of our knowledge, this work demonstrates for the first time that gradient-based HPO can be applied for long horizon problems like CIFAR-10 without being greedy. Specifically, we make the following contributions: **(1)** we propose to share hyperparameters through time and show that this significantly reduces the variance of hypergradients, **(2)** we show that the sign of hypergradients is a better indicator of convergence than their magnitude and enables a small number of outer steps, **(3)** we combine the above in a forward-mode algorithm adapted to modern SGD optimizers with momentum and weight decay, and **(4)** we show that our method significantly outperforms random search and greedy alternatives when used with the same computational budget.

## 2 RELATED WORK

There are many ways to perform hyperparameter optimization (HPO), including Bayesian optimization (Snoek et al., 2015), reinforcement learning (Zoph & Le, 2017), a mix of the two (Falkner et al., 2018), evolutionary algorithms (Jaderberg et al., 2017) and gradient-based methods (Bengio, 2000). Here we focus on the latter, but a comparison of HPO methods can be found in (Feurer & Hutter, 2019). Modern work in meta-learning deals with various forms of gradient-based HPO, many examples of which are discussed in this survey (Hospedales et al., 2020). However, meta-learning typically focuses on the few-shot regime where horizons are conveniently short, while in this work we focus on long horizons.

**Gradient-based HPO.** Using the gradient of some validation loss with respect to the hyperparameters is typically the preferred choice when the underlying optimization is differentiable. This is a type of bilevel optimization (Franceschi et al., 2018) which stems from earlier work on backpropagation through time (Werbos, 1990) and real-time recurrent learning (Williams & Zipser, 1989). Unfortunately, differentiating optimization is an expensive procedure in both time and memory, and most proposed methods are limited to small models and toy datasets (Domke, 2012; Maclaurin et al., 2015; Pedregosa, 2016). Efforts to make the problem more tractable include optimization shortcuts (Fu et al., 2016), truncation (Shaban et al., 2019) and implicit gradients (Rajeswaran et al., 2019; Lorraine et al., 2019). Truncation can be combined with our approach but produces biased gradients (Metz et al., 2019), while implicit differentiation is only applicable to hyperparameters that define the training loss (e.g. augmentation) but not to hyperparameters that define how the training loss is minimized (e.g. optimizer hyperparameters). Forward-mode differentiation (Williams & Zipser, 1989) boasts a memory cost constant with horizon size, but it doesn't address gradient degradation which has restricted its use to the greedy setting (Franceschi et al., 2017).

**Greedy Methods.** One trick that prevents gradient degradation and significantly reduces compute and memory cost is to solve the bilevel optimization greedily. This has become the default trick in various long-horizon problems, including HPO over optimizers (Luketina et al., 2016; Franceschi et al., 2017; Baydin et al., 2018; Donini et al., 2019), architecture search (Liu et al., 2019), dataset distillation (Wang et al., 2018) or curriculum learning (Ren et al., 2018). Greediness refers to finding the best hyperparameters locally rather than globally. In practice, it involves splitting the inner optimization problem into smaller chunks (often just one batch), and solving for hyperparameters over these smaller horizons instead; often in an online fashion. In this paper we expand upon previous observations (Wu et al., 2018) and take the view that greediness fundamentally solves for the wrong objective. Instead, the focus of our paper is to extend forward-mode differentiation methods to the non-greedy setting.

**Gradient Degradation.** Gradient degradation of some scalar w.r.t a parameter is a broad issue that arises when that parameter influences the scalar in a chaotic fashion, such as through long chains of nonlinear mappings. This manifests itself in HPO as vanishing or exploding hypergradients, due to low or high curvature components of the validation loss surface respectively. This leads to hypergradients with a large variance, which we denote as *hypervariance* (more in Section 5.1), and which prevents long-horizon optimization. This is usually observed in the context of recurrent neural networks (Bengio et al., 1993; 1994), but also in reinforcement learning (Parmas et al., 2018) and HPO (Maclaurin et al., 2015). Solutions like LSTMs (Hochreiter & Schmidhuber, 1997) and gradient clipping (Pascanu et al., 2013) have been proposed, but are respectively inapplicable and insufficient to our problem. Variational optimization (Metz et al., 2019) and preconditioning warp-layers (Flennerhag et al., 2020) can mitigate gradient degradation, but these methods are expensive in memory and therefore are limited to small architectures and/or a few hundred inner steps. In comparison, we differentiate over $\sim 10^4$ inner steps for Wide ResNets (Zagoruyko & Komodakis, 2016).

## 3 BACKGROUND

### 3.1 PROBLEM STATEMENT

Consider a neural network with weights $\boldsymbol{\theta}$, trained to minimize a loss $\mathcal{L}_{train}$ over a dataset $\mathcal{D}_{train}$, with a gradient-based optimizer $\Phi : \boldsymbol{\theta}_{t+1} = \Phi(\boldsymbol{\theta}_t(\boldsymbol{\lambda}_{[0:t-1]}), \boldsymbol{\lambda}_{[t]})$. Here $\boldsymbol{\lambda} \in \mathbb{R}^{K \times T}$ is a hyperparameter matrix, with $K$ the number of hyperparameters used per step (e.g. 2 if learning momentum and learning rate), and $T$ the total number of update steps. Throughout this paper we write column indices in brackets to differentiate them from a variable evaluated at time $t$.

We would like to find $\boldsymbol{\lambda}^*$ such that the minimizer of $\mathcal{L}_{train}$, namely $\boldsymbol{\theta}^* \simeq \boldsymbol{\theta}_T(\boldsymbol{\lambda})$, also minimizes some generalization loss $\mathcal{L}_{val}$ on a validation dataset $\mathcal{D}_{val}$. Bilevel optimization (Stackelberg, 1952) is the usual framework to express this problem:

$$\boldsymbol{\lambda}^* = \arg\min_{\boldsymbol{\lambda}} \mathcal{L}_{val}(\boldsymbol{\theta}_T(\boldsymbol{\lambda}), \mathcal{D}_{val}) \tag{1}$$

$$\text{subject to} \quad \boldsymbol{\theta}_T = \arg\min_{\boldsymbol{\theta}} \mathcal{L}_{train}(\boldsymbol{\theta}, \mathcal{D}_{train}) \tag{2}$$

$$\text{solved with} \quad \boldsymbol{\theta}_{t+1} = \Phi(\boldsymbol{\theta}_t(\boldsymbol{\lambda}_{[0:t-1]}), \boldsymbol{\lambda}_{[t]}) \tag{3}$$

The inner loop (lower level) in Eq 2 expresses a constraint on the outer loop (upper level) in Eq 1. In gradient-based HPO, our task is to compute the hypergradient $d\mathcal{L}_{val}/d\boldsymbol{\lambda}$ and update $\boldsymbol{\lambda}$ accordingly. The horizon $H$ refers to the number of update steps taken in the inner loop (to optimize $\boldsymbol{\theta}$) before one step is taken in the outer loop (to optimize $\boldsymbol{\lambda}$). When solving Eq 1 non-greedily we have $H = T$. However, most modern approaches are greedy: they rephrase the above problem into a sequence of several independent problems of smaller horizons, where $\boldsymbol{\lambda}^*_{[t:t+H]}$ is learned in the outer loop subject to an inner loop optimization from $\boldsymbol{\theta}_t$ to $\boldsymbol{\theta}_{t+H}$ with $H \ll T$. Despite its popularity, we argue against greediness in Section 4.

## 3.2 Forward-mode differentiation of Modern Optimizers

In modern gradient-based HPO, reverse-mode differentiation is used in the inner optimization problem (Eq 2) to optimize $\boldsymbol{\theta}$. However, the memory cost of using it for the outer optimization (Eq 1) is $\mathcal{O}(FH)$ where $F$ is the memory cost of one forward pass through the network (weights plus activations). This is extremely limiting: for large networks, only $H \sim 10$ could be solved with modern GPUs, while problems like CIFAR-10 require $H \sim 10^4$. Instead, we must use forward-mode differentiation, which scales in memory as $\mathcal{O}(DN)$, where $D$ is the number weights and $N$ is the number of learnable hyperparameters. The additional scaling with $N$ is a limitation in many applications, but sharing hyperparameters (Section 4) conveniently allows for smaller values of $N$.

In this section we give a background on forward-mode differentiation of hyperparameters, and extend it to the most popular optimizer, namely SGD with momentum and weight decay. To the best of our knowledge, previous work focuses on simpler versions of this optimizer, usually by removing momentum and weight decay, and only learns the learning rate, greedily.

First, we use the chain rule to write:

$$\frac{d\mathcal{L}_{val}}{d\boldsymbol{\lambda}} = \frac{\partial\mathcal{L}_{val}}{\partial\boldsymbol{\theta}_T}\frac{d\boldsymbol{\theta}_T}{d\boldsymbol{\lambda}} \tag{4}$$

Note that the direct gradient has been dropped since $\partial\mathcal{L}_{val}/\partial\boldsymbol{\lambda} = 0$ for optimizer hyperparameters. The first term on the RHS is trivial and can be obtained with reverse-mode differentiation as usual. The second term is more problematic because $\boldsymbol{\theta}_T = \boldsymbol{\theta}_T(\boldsymbol{\theta}_{T-1}(\boldsymbol{\theta}_{T-2}(...), \boldsymbol{\lambda}_{[T-2]}), \boldsymbol{\lambda}_{[T-1]})$. We use the chain rule again to calculate this term recursively:

$$\frac{d\boldsymbol{\theta}_t}{d\boldsymbol{\lambda}} = \frac{\partial\boldsymbol{\theta}_t}{\partial\boldsymbol{\theta}_{t-1}}\bigg|_{\boldsymbol{\lambda}} \frac{d\boldsymbol{\theta}_{t-1}}{d\boldsymbol{\lambda}} + \frac{\partial\boldsymbol{\theta}_t}{\partial\boldsymbol{\lambda}}\bigg|_{\boldsymbol{\theta}_{t-1}} \quad \text{which we write as} \quad \boldsymbol{Z}_t = \boldsymbol{A}_t\boldsymbol{Z}_{t-1} + \boldsymbol{B}_t \tag{5}$$

where $\boldsymbol{Z}_t \in \mathbb{R}^{D\times N}$, $\boldsymbol{A}_t \in \mathbb{R}^{D\times D}$ and $\boldsymbol{B}_t \in \mathbb{R}^{D\times N}$. We use Pytorch's update rule for SGD (Paszke et al., 2019), with learning rate $\alpha_t$, momentum $\beta_t$, weight decay $\mu_t$ and velocity $\boldsymbol{v}_t$:

$$\boldsymbol{\theta}_t = \Phi(\boldsymbol{\theta}_{t-1}) = \boldsymbol{\theta}_{t-1} - \alpha\boldsymbol{v}_t \quad \text{where} \quad \boldsymbol{v}_t = \beta_t\boldsymbol{v}_{t-1} + (\partial\mathcal{L}_{train}/\partial\boldsymbol{\theta}_{t-1}) + \mu_t\boldsymbol{\theta}_{t-1} \tag{6}$$

Now let us consider the case where we learn the learning rate schedule, namely $\boldsymbol{\lambda} = \boldsymbol{\alpha}$. If we use the update rule without momentum (Donini et al., 2019), $\boldsymbol{B}_t$ is conveniently sparse: it is a $D \times N$ matrix that only has one non-zero column corresponding to the hyperparameters used at step $t$, i.e. $\alpha_t$. However we introduce momentum and therefore the velocity depends on the hyperparameters of previous steps. A further recursive term $\boldsymbol{C}_t = (\partial\boldsymbol{v}_t/\partial\boldsymbol{\lambda})$ must now be considered to get exact hypergradients. Putting it together (see Appendix A) we obtain:

$$\begin{cases} \boldsymbol{A}_t^{\boldsymbol{\alpha}} = \mathbb{1}^{D\times D} - \alpha_t\left(\dfrac{\partial^2\mathcal{L}_{train}}{\partial\boldsymbol{\theta}_{t-1}^2} + \mu_t\mathbb{1}^{D\times D}\right) \\[2mm] \boldsymbol{B}_t^{\boldsymbol{\alpha}} = -\beta_t\alpha_t\boldsymbol{C}_{t-1}^{\boldsymbol{\alpha}} - \delta_t^{D\times N}\left(\beta_t\boldsymbol{v}_{t-1} + \dfrac{\partial\mathcal{L}_{train}}{\partial\boldsymbol{\theta}_{t-1}} + \mu_t\boldsymbol{\theta}_{t-1}\right) \\[2mm] \boldsymbol{C}_t^{\boldsymbol{\alpha}} = \beta_t\boldsymbol{C}_{t-1}^{\boldsymbol{\alpha}} + \left(\mu_t\mathbb{1}^{D\times D} + \dfrac{\partial^2\mathcal{L}_{train}}{\partial\boldsymbol{\theta}_{t-1}^2}\right)\boldsymbol{Z}_{t-1}^{\boldsymbol{\alpha}} \end{cases} \tag{7}$$

where $\delta_t^{D\times N}(\boldsymbol{q})$ turns a vector $\boldsymbol{q}$ of size $D$ into a zero matrix of size $D \times N$ where the column corresponding to the hyperparameter used at step $t$ is set to $\boldsymbol{q}$. A similar technique can be applied to momentum and weight decay to get $\boldsymbol{Z}_T^{\boldsymbol{\beta}}$ and $\boldsymbol{Z}_T^{\boldsymbol{\mu}}$ (Appendix A). The first-order approximation of the above system corresponds to setting the Hessian to zero, but in practice we find it too crude of an approximation for large horizons. All hypergradient derivations in this paper were thoroughly checked with finite differences. Finally, note that the approach above can be extended to any differentiable hyperparameter. In particular, other optimizers like Adam (Kingma & Ba, 2015) simply have different $\boldsymbol{Z}_t$, $\boldsymbol{A}_t$ and $\boldsymbol{B}_t$ matrices.

---

**Algorithm 1:** Non-greedy learning of the learning rate schedule $\boldsymbol{\alpha}$, using forward-mode differentiation with hyperparameter sharing and a sign-based outer optimizer.

---

**for** $o$ *in* $1, 2, ...$ **do**

    **initialize:** $\mathcal{D}_{train}, \mathcal{D}_{val}, \boldsymbol{\theta} \in \mathbb{R}^D, \boldsymbol{\alpha} = \mathbf{0}^{N_\alpha}, Z^{\boldsymbol{\alpha}} = \mathbf{0}^{D \times N_\alpha}, C^{\boldsymbol{\alpha}} = \mathbf{0}^{D \times N_\alpha}, \boldsymbol{\gamma}^{\boldsymbol{\alpha}} \in \mathbb{R}^{N_\alpha}$

    **for** $i$ *in* $1, 2, ..., H$ **do**

        $\boldsymbol{x}_{train}, \boldsymbol{y}_{train} \sim \mathcal{D}_{train}$

        $\boldsymbol{g}_{train} = \partial \mathcal{L}_{train}(\boldsymbol{x}_{train}, \boldsymbol{y}_{train})/\partial \boldsymbol{\theta}$

        **if** *truncate==False* **then**

            $\mathcal{H}Z^{\boldsymbol{\alpha}} = \partial(\boldsymbol{g}_{train} Z^{\boldsymbol{\alpha}})/\partial \boldsymbol{\theta}$

            $Z^{\boldsymbol{\alpha}} = A^{\boldsymbol{\alpha}} Z^{\boldsymbol{\alpha}} + B^{\boldsymbol{\alpha}}$    (Eq 7)

        update $\boldsymbol{\theta}$ and $\boldsymbol{v}$    (Eq 6)

    $\boldsymbol{g}_{val} = \partial \mathcal{L}_{val}(\mathcal{D}_{val})/\partial \boldsymbol{\theta}$

    $\boldsymbol{s}_o = \text{sgn}(\boldsymbol{g}_{val} Z^{\boldsymbol{\alpha}})$                               `// hypergradient signs`

    **for** $n$ *in* $1, 2, ..., N_\alpha$ **do**

        **if** $\boldsymbol{s}_{o,[n]} \neq \boldsymbol{s}_{o-1,[n]}$ **then**

            $\boldsymbol{\gamma}_{[n]}^{\boldsymbol{\alpha}} \leftarrow \boldsymbol{\gamma}_{[n]}^{\boldsymbol{\alpha}}/2$

    $\boldsymbol{\alpha} \leftarrow \boldsymbol{\alpha} - \boldsymbol{s}_o \odot \boldsymbol{\gamma}^{\boldsymbol{\alpha}}$

---

## 4 ENABLING NON-GREEDINESS

Greedy approaches (Luketina et al., 2016; Franceschi et al., 2017; Baydin et al., 2018; Donini et al., 2019) have many advantages: they mitigate gradient degradation issues, lower training time, and require less memory in reverse-mode differentiation. However, these methods look for $\boldsymbol{\lambda}^*$ that does well locally rather than globally, i.e. they constrain $\boldsymbol{\lambda}^*$ to a subspace of solutions such that $\boldsymbol{\theta}_H^*, \boldsymbol{\theta}_{2H}^*, ..., \boldsymbol{\theta}_T^*$ all yield good validation performances. In the same way that running a marathon by optimizing for a sequence of 100m sprints is sub optimal, greediness is a poor proxy to minimize $\mathcal{L}_{val}(\boldsymbol{\theta}_T)$. In our experiments, we found that getting competitive hyperparameters with greediness often revolves around tricks such as online learning with a very low outer learning rate combined with hand-tuned initial hyperparameter values, to manually prevent convergence to small values. But solving the greedy objective correctly leads to poor solutions, a special case of which was previously described as the "short-horizon bias" (Wu et al., 2018) when learning the learning rate. Finally, we found that most of the greedy literature uses the test set as the validation set, which creates a risk of meta-overfitting to the test set. Throughout this work, we carve out a validation set from our training set instead.

The main challenge of doing non-greedy bilevel optimization over long horizons is gradient degradation. In HPO this arises because a small change in the state of the network at inner step $t$ can cascade forward into a completely different weight trajectory given enough subsequent steps, yielding very different hypergradients. We refer to this phenomenon as *hypervariance*, and quantify it in Section 5.1.

**Hyperparameter Sharing.** One way to iron out this hypervariance would be to take many outer steps on the hyperparameters, with previous work suggesting that a few thousands would be required (Wu et al., 2018). But since each outer step requires computing the entire inner problem, this is intractable. Instead, we propose to use the long horizon to our advantage and *average out hypergradients through inner steps rather than outer steps*. This is equivalent to hyperparameter sharing: learning one hyperparameter for several contiguous steps in the inner loop. Assuming that contiguous steps have hypergradients drawn from the same distribution, the average hypergradient is more stable, since it is less likely to reflect a single batch in the inner loop. The key to this solution

being effective is that it gets better as degradation gets worse: the larger the horizon, the more noisy individual hypergradients become, but the more samples are included in the average hypergradient, and so the less noisy the average hypergradient becomes. Assuming $n$ contiguous hypergradients are drawn from the same distribution, the standard error of their mean will scale as $1/\sqrt{n}$.

**Convergence From Hypergradient Signs.** While hyperparameter sharing produces stable hypergradients with a reasonable range (see Appendix C), reaching zero hypergradients and thus convergence remains difficult, and often requires a large number of outer steps. We propose to use the sign of hypergradients as an indicator of convergence instead of their magnitude. More specifically, we update hyperparameters with positive (negative) hypergradients by an amount $+\gamma$ $(-\gamma)$, and $\gamma$ is decayed by a factor of 2 every time the corresponding hypergradient changes sign across consecutive outer steps. This allows for convergence after hypergradients have changed signs a few times, and we find this much more robust than using modern optimizers to update $\boldsymbol{\lambda}$ like Adam. This has the added benefit of letting the user define the range of hyperparameter search more explicitly, rather than having it implicitly defined by the magnitude of hypergradients.

Putting it all together, Algorithm 1 shows how to learn a schedule of $N^{\boldsymbol{\alpha}}$ learning rates efficiently. Hyperparameter sharing is implemented by reusing the same $Z^{\boldsymbol{\alpha}}$ columns for several consecutive inner steps. The most expensive part per outer step is to compute the Hessian matrix product $\mathcal{H}Z^{\boldsymbol{\alpha}}$ at each inner step $i$. We can adapt the work of (Shaban et al., 2019) to forward mode differentiation by truncating alternating steps. We could also use functional forms for more complex schedules to be learned in terms of fewer hyperparameters, which would cheapen the calculation of $\mathcal{H}Z^{\boldsymbol{\alpha}}$. However, this typically results in including a stronger inductive bias about the general shape of each hyperparameter schedule, which can easily cloud the true performance of HPO algorithms. In practice, we use a similar form to Algorithm 1 to learn the learning rate, momentum and weight decay jointly.

## 5 EXPERIMENTS

In Section 5.1 we quantify gradient degradation and show that non-greediness increases it while hyperparameter sharing significantly reduces it. In Section 5.2 we show that our method allows for stable non-greedy HPO on 3 datasets, and outperforms several other methods when used with the same computational budget. Implementation details for each of our experiments can be found in Appendix B.

### 5.1 REDUCING GRADIENT DEGRADATION

Gradient degradation in the context of HPO over long horizons isn't well understood. In this section we propose a simple metric to quantify this degradation (hypervariance), and we propose a method to reduce it (hyperparameter sharing). We also consider the fluctuation of the sign of the hypergradients to motivate the use of the outer optimizer described in Section 4.

**Hypervariance Definition.** Consider the hypergradients of the learning rates $\boldsymbol{\alpha}$. These are affected by four main factors: the training data, the validation data, the weights' initialization $\boldsymbol{\theta}_0$ and the current learning rate schedule $\boldsymbol{\alpha}_0$. We can slightly perturb each factor individually $P$ times while keeping all others constant. This gives us a distribution of hypergradients around a fixed point for each hyperparameter, which should have low variance in order for the outer optimization to be stable. We define the hypervariance to be a (unitless) inverse signal to noise ratio, namely the ratio of the standard deviation to the mean of hypergradients, computed for each hyperparameter. This makes the norm of the hypergradients irrelevant, which is desired since it can always be scaled as needed with some outer learning rate in practice.

**Hypergradient Sign Fluctuation.** In Section 4 we made the case for using the sign of hypergradients alone. Since hypervariance doesn't capture the fluctuation in the sign of hypergradients, we also measure this quantity by looking at the smallest quantity between $P_+$ and $P_-$, the number of positive and negative signs, expressed as a ratio: sign fluctuation $= \min(P_-/P, P_+/P)$. This quantity is capped at $50\%$ which corresponds to hypergradients maximally uninformative regarding whether or not some hyperparameter should increase or decrease.

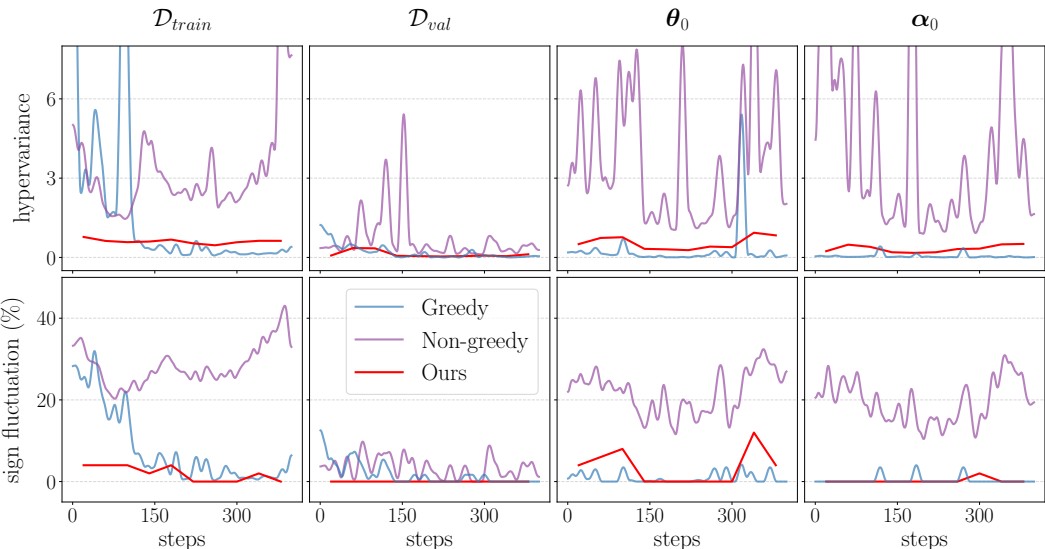

Figure 2: The hypervariance (top row) and sign fluctuation (bottom row), when learning several learning rates on the SVHN dataset. These are calculated as we perturb, from left to right, the choice of training data, the choice of validation data, the initial weights and the initial learning rates. We observe that non-greediness over long horizons is responsible for a large hypervariance and sign fluctuation (purple vs blue). However, we can preserve non-greediness and greatly reduce gradient degradation by sharing hyperparameters for each 40 contiguous inner steps (red).

The variance and sign fluctuation of hypergradients are shown in Figure 2 when hypergradients are calculated for several learning rates, for a LeNet on the SVHN dataset. We observe that non-greediness makes hypergradients much more sensitive to small changes in the optimization landscape. For instance, changing the seed for the sampling of the training dataset produces hypergradients with a standard deviation that is over 6 times larger than the mean, making outer optimization very inefficient. However, sharing contiguous hyperparameters lowers the hypervariance and sign fluctuation drastically. In particular, the sign of hypergradients only fluctuates by an average of 1.5% across all factors of variation, making it a robust quantity for outer optimization.

## 5.2 NON-GREEDY HPO OVER LONG HORIZONS

The section above considered hypergradient noise around a fixed hyperparameter setting. In this section, we consider how that noise and its mitigation translate to the outer optimization process, where hyperparameters are updated at each outer step.

**MNIST & SVHN.** We start with HPO over medium length horizons with small networks, where reverse-mode differentiation can be used to learn many hyperparameters non-greedily as a baseline. In Figure 3, we learn the learning rate schedule starting from $\alpha = 0$, in the maximally greedy and non-greedy setting, with and without hyperparameter sharing. We do not use tricks like online learning as in (Baydin et al., 2018) to make greediness more transparent. As previously observed by Wu et al. (2018), greedy optimization leads to poor solutions with learning rates that are too small. While the vanilla non-greedy setting works well for simple datasets like MNIST, it fails to learn a good schedule for real-world datasets like SVHN, converging to much higher values than reasonable. By sharing hyperparameters, we stabilize the outer optimization by lowering hypervariance, and this allows us to learn a schedule that even beats an off-the-shelf baseline for this dataset.

**CIFAR-10.** We test Algorithm 1 on 50 epochs of CIFAR-10, for a Wide ResNet of 16 layers. Results are shown in Figure 1. We choose not to use larger architectures or more epochs to save compute time, and because we find that hyperparameters matter most for fewer epochs. Note also that we are not interested in the absolute performance of the model, but rather in the performance of the hyperparameters for any given model. To the best of our knowledge, we are the first to

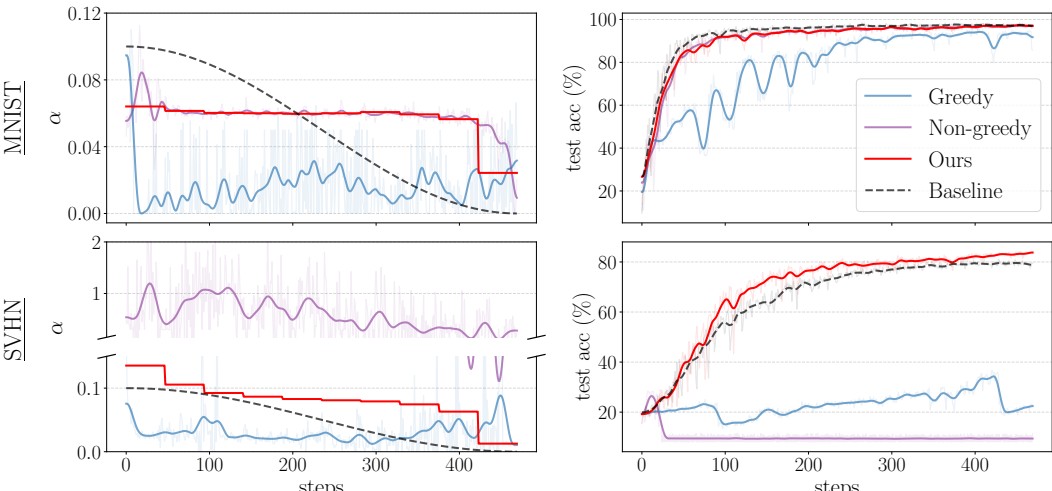

Figure 3: The learning rate schedule $\alpha$ learned for the MNIST and SVHN datasets using a LeNet architecture over 500 inner gradient steps. We observe that on real-world datasets like SVHN, both greedy and non-greedy hyperparameter optimizations fail to learn decent learning rate schedules. However, sharing learning rates within neighbouring batches stabilizes non-greedy hypergradients and allows us to find schedules that can even outperform off-the-shelf schedules.

demonstrate a stable gradient-based optimization of the learning rate, momentum and weight decay for such a large number of steps ($\sim 10^4$). Interestingly, our algorithm mostly relies on high weight decays for regularization, which allows for smaller momentum with no performance drop. Finally, we note that the search range of our algorithm is much larger than typical methods learning similar hyperparameters. By initializing $\alpha$ to zero and $\gamma^\alpha$ to 0.1, taking 10 outer steps effectively defines a search range of $[-1, 1]$ which is about 20 times larger than usual competing methods on the same dataset (Falkner et al., 2018). In short, we assume much less prior knowledge about the nature of the hyperparameters to learn. This makes it much harder for non-gradient based methods like random search or Bayesian optimization to find good hyperparameter configurations.

**Baseline Comparison.** We compare our algorithm to baselines in Table 1. A common theme in meta-learning research has been the lack of appropriate baselines, with researchers often finding that random search (RS) can outperform complex search algorithms, for instance in NAS (Li & Talwalkar, 2019) or automatic augmentation (Cubuk et al., 2020). One reason why RS may beat greedy alternatives is precisely that RS isn't greedy. Another reason is that many of these applications define a search range that is contrived to have mostly good candidates. In the case of DARTS (Liu et al., 2019), expanding the search space to include many poor architectures has helped diagnose issues with its search algorithm (Zela et al., 2020). We run RS over the same search space as our method, and for a comparable GPU time ($\sim 13$ hours). As stated above, our search space is large and this makes it difficult for RS to find good hyperparameters quickly. Bayesian optimization underperforms RS in our experimental setting, and we refer the reader to Appendix E for more details. We also evaluate hypergradient descent (Baydin et al., 2018) with various initial hyperparameters sampled from our search range, so as to match its computational budget with ours.

**Hand-tuned Comparison.** Finally, we manually search for the best hyperparameters by using the ones that are most common in the literature, and evaluating multiple values around them (see Appendix D). Note that this manual baseline relies on strong inductive biases which wouldn't be available, say, when learning hyperparameters for a new unknown optimizer. Our method significantly outperforms both random search and hypergradient descent, while matching the performance of the best hand-tuned hyperparameters. The latter suggests that our method would be an efficient way to perform HPO for new applications or new optimizers, where good hyperparameter priors don't already exist.

Table 1: Test accuracy of our method compared to baselines ran for the same computational budget. MNIST and SVHN are optimized over 2 epochs ($\sim 10^3$ steps) while CIFAR-10 is optimized over 50 epochs ($\sim 10^4$ steps). Note how greediness gets worse as the dataset becomes more complex.

| Method | MNIST | SVHN | CIFAR-10 |
|---|---|---|---|
| Hand-tuned (best choice) | $98.4_{\pm 0.1}$ | $85.1_{\pm 0.3}$ | $89.2_{\pm 0.2}$ |
| Random Search | $98.0_{\pm 0.2}$ | $82.2_{\pm 1.1}$ | $82.3_{\pm 0.7}$ |
| Greedy (Baydin et al., 2018) | $97.5_{\pm 0.1}$ | $84.0_{\pm 0.2}$ | $80.8_{\pm 0.2}$ |
| Ours | $\mathbf{98.7}_{\pm 0.1}$ | $\mathbf{85.7}_{\pm 0.8}$ | $\mathbf{89.3}_{\pm 0.1}$ |

## 6 DISCUSSION

This work makes an important step towards gradient-based HPO for long horizons, by enabling non-greediness through the mitigation of gradient degradation. More specifically, we show that sharing contiguous hyperparameters is a simple and efficient way to reduce the variance of hypergradients. The nature of non-greedy bilevel optimization requires the inner optimization to be run several times, and so HPO over ImageNet-like datasets remains costly. However, we show that only $\sim 10$ outer steps are sufficient to converge to good hyperparameters, as opposed to the thousands conjectured in previous work (Wu et al., 2018). This is in part made possible by high quality hypergradients, and by using the fluctuation of their sign as a criterion for convergence. Extensions to this work could consider extending our method to discrete-valued hyperparameters with relaxation techniques, which would enable a larger range of non-greedy meta-learning applications. We hope that our work encourages the community to reconsider gradient-based hyperparameter optimization in terms of non-greediness, and pave the way towards a ubiquitous hyperparameter solver.

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

# Appendices

## A: Forward-mode Hypergradient Derivations

Recall that we are interested in calculating

$$\boldsymbol{Z}_t = \boldsymbol{A}_t \boldsymbol{Z}_{t-1} + \boldsymbol{B}_t$$

recursively during the inner loop, where

$$\boldsymbol{Z}_t = \frac{d\boldsymbol{\theta}_t}{d\boldsymbol{\lambda}} \qquad \boldsymbol{A}_t = \left.\frac{\partial \boldsymbol{\theta}_t}{\partial \boldsymbol{\theta}_{t-1}}\right|_{\boldsymbol{\lambda}} \qquad \boldsymbol{B}_t = \left.\frac{\partial \boldsymbol{\theta}_t}{\partial \boldsymbol{\lambda}}\right|_{\boldsymbol{\theta}_{t-1}}$$

so that we can calculate the hypergradients on the final step using

$$\frac{d\mathcal{L}_{val}}{d\boldsymbol{\lambda}} = \frac{\partial \mathcal{L}_{val}}{\partial \boldsymbol{\theta}_T} \boldsymbol{Z}_T$$

Each type of hyperparameter needs its own matrix $\boldsymbol{Z}_t$, and therefore its own matrices $\boldsymbol{A}_t$, and $\boldsymbol{B}_t$.

Consider first the derivation of these matrices for the learning rate, namely $\boldsymbol{\lambda} = \boldsymbol{\alpha}$. Recall that the update rule after substituting the velocity $\boldsymbol{v}_t$ in is

$$\boldsymbol{\theta}_t = \boldsymbol{\theta}_{t-1} - \alpha_t \left( \beta_t \boldsymbol{v}_{t-1} + \frac{\partial \mathcal{L}_{train}}{\partial \boldsymbol{\theta}_{t-1}} + \mu_t \boldsymbol{\theta}_{t-1} \right)$$

and therefore it follows directly that

$$\boldsymbol{A}_t^{\boldsymbol{\alpha}} = \mathbb{1}^{D \times D} - \alpha_t \left( \frac{\partial^2 \mathcal{L}_{train}}{\partial \boldsymbol{\theta}_{t-1}^2} + \mu_t \mathbb{1}^{D \times D} \right)$$

The calculation of $\boldsymbol{B}_t^{\boldsymbol{\alpha}}$ is a bit more involved in our work because when using momentum $\boldsymbol{v}_{t-1}$ is now itself a function of $\boldsymbol{\alpha}$. First we write

$$\boldsymbol{B}_t^{\boldsymbol{\alpha}} = -\beta_t \left( \frac{\partial \alpha_t}{\partial \boldsymbol{\alpha}} \boldsymbol{v}_{t-1} + \alpha_t \frac{\partial \boldsymbol{v}_{t-1}}{\partial \boldsymbol{\alpha}} \right) - \frac{\partial \alpha_t}{\partial \boldsymbol{\alpha}} \left( \frac{\partial \mathcal{L}_{train}}{\partial \boldsymbol{\theta}_{t-1}} + \mu_t \boldsymbol{\theta}_{t-1} \right)$$

$$= -\beta_t \alpha_t \frac{\partial \boldsymbol{v}_{t-1}}{\partial \boldsymbol{\alpha}} - \delta_t^{D \times N} \left( \beta_t \boldsymbol{v}_{t-1} + \frac{\partial \mathcal{L}_{train}}{\partial \boldsymbol{\theta}_{t-1}} + \mu_t \boldsymbol{\theta}_{t-1} \right)$$

Now since

$$\boldsymbol{v}_t = \beta_t \boldsymbol{v}_{t-1} + \frac{\partial \mathcal{L}_{train}}{\boldsymbol{\theta}_{t-1}} + \mu_t \boldsymbol{\theta}_{t-1}$$

we can write the partial derivative of the velocity as an another recursive rule:

$$\boldsymbol{C}_t^{\boldsymbol{\alpha}} = \frac{\partial \boldsymbol{v}_t}{\partial \boldsymbol{\alpha}}$$

$$= \beta_t \boldsymbol{C}_{t-1}^{\boldsymbol{\alpha}} + \frac{\partial^2 \mathcal{L}_{train}}{\partial \boldsymbol{\alpha} \partial \boldsymbol{\theta}_{t-1}} + \mu_t \frac{\partial \boldsymbol{\theta}_{t-1}}{\partial \boldsymbol{\alpha}}$$

$$= \beta_t \boldsymbol{C}_{t-1}^{\boldsymbol{\alpha}} + \left( \mu_t \mathbb{1}^{D \times D} + \frac{\partial^2 \mathcal{L}_{train}}{\partial \boldsymbol{\theta}_{t-1}^2} \right) \frac{\partial \boldsymbol{\theta}_{t-1}}{\partial \boldsymbol{\alpha}}$$

And putting all together recovers the system:

$$
\begin{cases}
\boldsymbol{A}_t^{\boldsymbol{\alpha}} = \mathbb{1}^{D \times D} - \alpha_t \left( \dfrac{\partial^2 \mathcal{L}_{train}}{\partial \boldsymbol{\theta}_{t-1}^2} + \mu_t \mathbb{1}^{D \times D} \right) \\[2ex]
\boldsymbol{B}_t^{\boldsymbol{\alpha}} = -\beta_t \alpha_t \boldsymbol{C}_{t-1}^{\boldsymbol{\alpha}} - \delta_t^{D \times N} \left( \beta_t \boldsymbol{v}_{t-1} + \dfrac{\partial \mathcal{L}_{train}}{\partial \boldsymbol{\theta}_{t-1}} + \mu_t \boldsymbol{\theta}_{t-1} \right) \\[2ex]
\boldsymbol{C}_t^{\boldsymbol{\alpha}} = \beta_t \boldsymbol{C}_{t-1}^{\boldsymbol{\alpha}} + \left( \mu_t \mathbb{1}^{D \times D} + \dfrac{\partial^2 \mathcal{L}_{train}}{\partial \boldsymbol{\theta}_{t-1}^2} \right) \boldsymbol{Z}_{t-1}^{\boldsymbol{\alpha}}
\end{cases}
$$

For learning the momentum and weight decay, a very similar approach yields

$$
\begin{cases}
\boldsymbol{A}_t^{\boldsymbol{\beta}} = \mathbb{1}^{D \times D} - \alpha_t \left( \dfrac{\partial^2 \mathcal{L}_{train}}{\partial \boldsymbol{\theta}_{t-1}^2} + \mu_t \mathbb{1}^{D \times D} \right) \\[2ex]
\boldsymbol{B}_t^{\boldsymbol{\beta}} = -\beta_t \alpha_t \boldsymbol{C}_{t-1}^{\boldsymbol{\beta}} - \delta_t^{D \times N} (\alpha_t \boldsymbol{v}_{t-1}) \\[2ex]
\boldsymbol{C}_t^{\boldsymbol{\beta}} = \delta_t^{D \times N}(\boldsymbol{v}_t) + \beta_t \boldsymbol{C}_{t-1}^{\boldsymbol{\beta}} + \left( \mu_t \mathbb{1}^{D \times D} + \dfrac{\partial^2 \mathcal{L}_{train}}{\partial \boldsymbol{\theta}_{t-1}^2} \right) \boldsymbol{Z}_{t-1}^{\boldsymbol{\beta}}
\end{cases}
$$

and

$$
\begin{cases}
\boldsymbol{A}_t^{\boldsymbol{\mu}} = \mathbb{1}^{D \times D} - \alpha_t \left( \dfrac{\partial^2 \mathcal{L}_{train}}{\partial \boldsymbol{\theta}_{t-1}^2} + \mu_t \mathbb{1}^{D \times D} \right) \\[2ex]
\boldsymbol{B}_t^{\boldsymbol{\mu}} = -\beta_t \alpha_t \boldsymbol{C}_{t-1}^{\boldsymbol{\mu}} - \delta_t^{D \times N} (\alpha_t \boldsymbol{\theta}_{t-1}) \\[2ex]
\boldsymbol{C}_t^{\boldsymbol{\mu}} = \delta_t^{D \times N}(\boldsymbol{\theta}_{t-1}) + \beta_t \boldsymbol{C}_{t-1}^{\boldsymbol{\mu}} + \left( \mu_t \mathbb{1}^{D \times D} + \dfrac{\partial^2 \mathcal{L}_{train}}{\partial \boldsymbol{\theta}_{t-1}^2} \right) \boldsymbol{Z}_{t-1}^{\boldsymbol{\mu}}
\end{cases}
$$

## B: Implementation Details

**Figure 1.** We learn $5$ values for the learning rates, $1$ for the momentum and $1$ for the weight decay, to make it comparable to the hyperparameters used in the literature for CIFAR-10 (see Appendix D). A batch size $256$ is used, with $5\%$ of the training set of each epoch set aside for validation. We found larger validation sizes not to be helpful. Hypergradient descent uses hyperparameters initialized at zero as well, and trains all hyperparameters online with an SGD outer optimizer with learning rate $0.2$ and $\pm 1$ clipping of the hypergradients. We used initial values $\gamma_\alpha = 0.1, \gamma_\beta = 0.15$ and $\gamma_\mu = 4 \times 10^{-4}$ but the performance barely changed when these values were multiplied or divided by $2$. Since we take $10$ outer steps and initialize all hyperparameters at zero, this defines a search ranges: $\alpha \in [-1, 1]$, $\beta \in [-1.5, 1.5]$, and $\gamma \in [-4 \times 10^{-3}, 4 \times 10^{-3}]$. The Hessian matrix product is clipped to $\pm 10$ to prevent one batch from having a dominating contribution to hypergradients. Strictly speaking, sharing hyperparameters corresponds to summing their hypergradients rather than averaging them, and therefore we divide this sum by the number of batches each hyperparameter is used for.

**Figure 2.** Here we wanted to isolate all factors responsible for hypervariance. We thus used float64 precision with batch size $64$, as this reduced hypervariance across all methods. Clipping did not change the hypervariance drastically but was applied to $\pm 1$ in the inner loop. We learned $10$ learning rates for our method, namely one learning rate per $40$ steps. The unperturbed learning rate was set to $\boldsymbol{\alpha}_0 = 0.05$. The perturbation for initial weights and hyperparameters (here $\boldsymbol{\alpha}$) corresponds to adding $\pm 1\%$ of their value, while perturbation of $\mathcal{D}_{train}$ and $\mathcal{D}_{val}$ correspond to a different sampling seed. For the plots, we used a moving average for the greedy and non-greedy lines for clarity.

**Figure 3.** Here we used a batch size of $128$ for both datasets to allow $2$ epochs worth of inner optimization in about $500$ inner steps. Clipping was restricted to $\pm 3$ to show the effect of noisy hypergradients more clearly. Since MNIST and SVHN are cheap datasets to run on a LeNet architecture, we can afford $50$ outer steps and early stopping based on validation accuracy.

**Table 1.** Each method for a given dataset is run for a similar GPU budget. For CIFAR-10 that's about 13 hours on a 2080 GTX GPU. Random search and our method both consider 7 learning rates, 1 momentum and 1 weight decay. Note that random search struggles in finding good hyperparameters even though $\sim 100$ random hyperparameter settings are evaluated, because some hyperparameters can compromise the whole training process. This is the case of large positive learning rates, negative learning rates, or momentums greater than one. Since random search is a memory-less trial-and-error method, any hyperparameter region that compromises learning altogether is very harmful. Truncation was kept $\leq 15\%$ to obtain maximal performance. The error bars are calculated over 3 seeds for each entry in the table.

## C: Hypergradients

Here we provide the raw hypergradients corresponding to the outer optimization shown in Figure 1. Note that the range of these hypergradients is made reasonable by the averaging of gradients coming from contiguous hyperparameters.

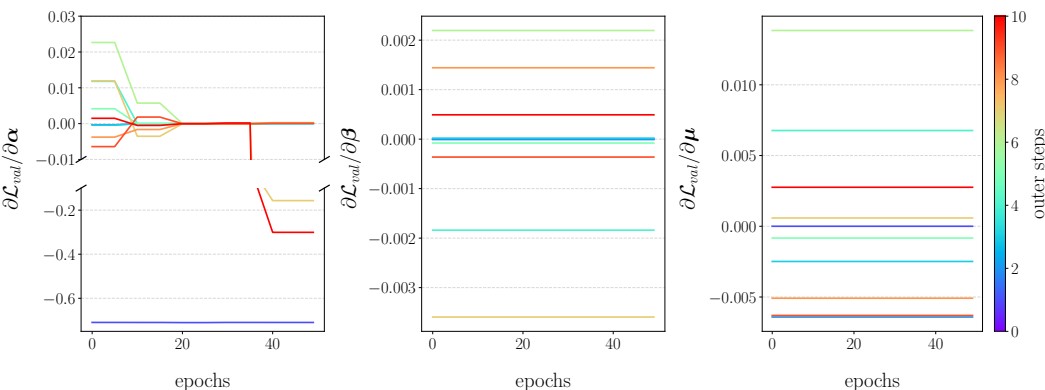

Figure 4: Hypergradients have a reasonable range but fail to always converge to zero when the validation performance stops improving.

## D: Baselines

The objective here is to select the best hyperparameter setting that a deep learning practitioner would reasonably be expected to use, based on the hyperparameters used by the community for the datasets at hand. For CIFAR-10, the most common hyperparameter setting is the following: $\alpha$ is initialized at $\alpha_0 = 0.2$ (for batch size 256, as used in our experiments) and decayed by a factor $\eta = 0.2$ at $30\%, 60\%$ and $80\%$ of the run (`MultiStep` in Pytorch); the momentum $\beta$ is constant at $0.9$, and the weight decay $\mu$ is constant at $5 \times 10^{-4}$. We search for combinations of hyperparameters around this setting. More specifically, we search over all combinations of $\alpha_0 = \{0.05, 0.1, 0.2, 0.4, 0.6\}$, $\eta = \{0.1, 0.2, 0.4\}$, $\beta = \{0.45, 0.9, 0.99\}$, and $\mu = \{2.5 \times 10^{-4}, 5 \times 10^{-4}, 1 \times 10^{-3}\}$. This makes up a total of 135 hyperparameter settings, which we each run 3 times to get a mean and standard deviation. The distribution of those means are provided in Figure 5, and the best hyperparameter setting is picked based on validation performance. This is the value we report in Table 1 under *Hand-tuned (best)*.

MNIST and SVHN hyperparameters matter less, and in particular we observed no gain from using momentum and weight decay. The most popular learning rate schedules used for these datasets seem to be the cosine annealing one. We evaluate this schedule for $\alpha_0 = \{0.05, 0.1, 0.2, 0.4, 0.6\}$ and select the best hyperparameters based on validation performance.

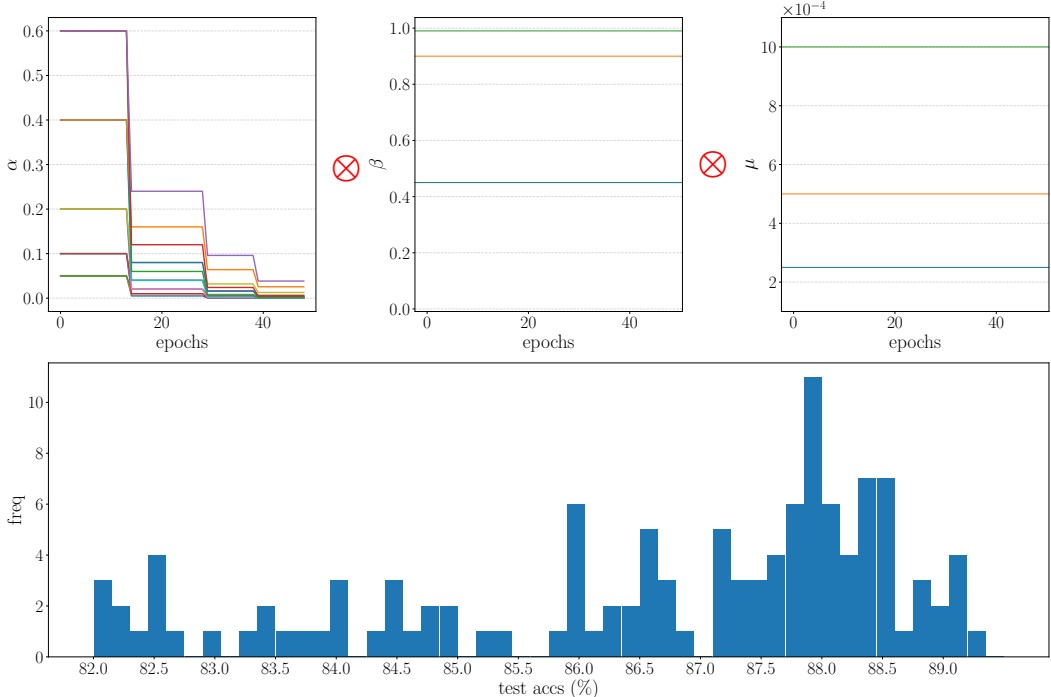

Figure 5: The combination of hyperparameters searched over for CIFAR-10 (top row) and the corresponding distribution of test accuracies (bottom row).

## E: Bayesian Optimization

We ran numerous BO experiments for our experimental configuration. We used the modern BoTorch-Ax library (Balandat et al., 2019), with the common BO settings found in the HPO literature: Gaussian process with Matern 5/2 kernel, with expected improvement (EI) as the acquisition function, and using Sobol search for the first few arms.

In our experiments, we sought to showcase the efficiency of using gradients in HPO by considering a large hyperparameter search range. This means that we assume very little about the hyperparameters we are trying to learn. For instance we learn the learning rate in a range $\alpha \in [-1, 1]$ while a common search range in BO would be about 20 times smaller like $\alpha \in [10^{-6}, 10^{-1}]$ (Falkner et al., 2018), and would use log scaling to search this space more efficiently. In our setting, we find that BO and related methods struggle to even match random search. This is because the search space mostly contains configurations that have a high validation loss, and BO spends much of its time exploring irrelevant fluctuations in that high loss region. In order to slightly outperform random search with BO, we had to divide the search range by $\sim 20$ and use 5 times the computational budget. Indeed, BO is usually orders of magnitude more expensive to run than our method, with some methods reporting up to 33 GPU days on CIFAR-10 (Falkner et al., 2018) while our algorithm takes $\sim 13$ hours (albeit for smaller architectures).

