# OpenReview forum: "Non-greedy Gradient-based Hyperparameter Optimization Over Long Horizons"
_ICLR.cc/2021/Conference — Reject_

### Official Review · AnonReviewer1 · 2020-10-25
**This paper tackles the important problem of HPO, and focus on the optimization of real-valued hyperparameters for DNNs using a gradient-based method**

**Rating:** 7
**Confidence:** 2

**Review:**

The authors tackle the problem of HPO, focusing on the optimization of real-valued hyperparameters for DNNs using a gradient-based method. This novel method enables non-greediness through the mitigation of gradient degradation and allows for long horizons.

How does this work compare against tools like BOHB (not cited in this paper) [1]? The method introduced here is based on hypergradients while BOHB is based on a combination of Hyperband and Bayesian optimization, so these are two different algorithms. Nonetheless, since BOHB can perhaps be considered state-of-the-art for DNN HPO including continuous and discrete hyperparameters, it is still worth to provide a  comparison in the experimental section. This is reinforced by the fact that the authors use random sampling as a baseline which has been shown to be inferior to BOHB on similar or identical benchmarks.

In the methodology section, 3.1 and 3.2 are background material. It'd perhaps be clearer if the authors move these parts in a background section so as to differentiate the contributions from the background. The contribution section of this paper is section 3.3 where the key idea is the reduction of the hypervariance induced by the gradient degradation phenomenon. In the experiment section, the authors define the hypervariance, which should be defined in a previous section instead of the empirical part. Or else, since the hypervariance is somewhat defined twice, these two parts could be merged in section 3.3.

To reduce the hypervariance the authors propose hyperparameter sharing and show empirically how it helps gradient degradation. In addition, the authors propose to use gradient signs as an indicator of convergence and show the impact empirically. No formal justification is provided for these two contributions which is a weakness of the paper.

Minor remarks:
- The abstract should explicitly mention that this is a Deep learning paper. The authors shouldn't expect that since the paper is submitted at ICLR then a reader will know that the paper is on DL.
- It is usually a good idea to have the figures either at the beginning or at the end of a page rather than in the middle of the page. See for example Figure 2.
- It is odd to have figure 1 on the second page if the figure is then used on page 8. On page 2, the readers are not able to understand the figure because the outer step in the figure is introduced later in the text.

[1] Falkner, Stefan, Aaron Klein, and Frank Hutter. "BOHB: Robust and Efficient Hyperparameter Optimization at Scale." ICML. 2018.

---

> ### Author Response · Authors · 2020-11-18
> **Reply to reviewer 1**
>
> Thank you for your thorough comments and encouraging review. We hope to address your points in what follows.
>
>
> 1. *How does this work compare against tools like BOHB?* We have included a separate comment on experiments based on Bayesian optimization on this page. The bottom line is that methods that rely on Bayesian optimization (like BOHB) struggle to even match random search in our experimental setting, where the search range is large and the computational budget is low. For instance, BOHB searches hyperparameters in narrow ranges that only contain reasonably good values (e.g $\alpha \in [10^{-6}, 10^{-1}]$ using a log scale inductive bias) while our experiment makes much fewer assumptions about the hyperparameters we are searching  ($\alpha \in [-1, 1]$). Finally, our method runs in about 13 GPU hours while BOHB is run for 33 GPU days (albeit on a larger architecture). We have added the BOHB paper in related work, because it is still relevant to our paper.
>
>
> 2. *Structure of the paper.* We agree that a Background section would make our contribution clearer, so we have moved 3.1 and 3.2 to section "3 Background" and changed 3.3 to section "4.0 Enabling Non-Greediness". Thank you.
>
>
> 3. *Justification for hyperparameter sharing and sign-based outer optimizer.* While we concede that a more theoretical approach could make up interesting future work, there is some justification for these heuristics. Hyperparameter sharing averages the hypergradient of contiguous hyperparameters, and therefore is expected to reduce noise, because the standard error of the mean (SEM) is inversely proportional to $\sqrt{n}$, where $n$ is the number of samples to calculate the mean (here, the number of shared steps). Of course this picture only holds if we assume that the hypergradients for contiguous steps come from the same distribution, which may not be the case. This point was added in section 4. Finally, using the oscillation in sign as a measure of convergence is motivated by the following. If the hypergradient is negative at $\alpha_1$, the hyperparameter is updated to $\alpha_2 = \alpha_1 + \gamma$ with $\gamma > 0$. Now if the hypergradient at $\alpha_2$ is positive, it is a signal that the best $\alpha$ hyperparameter is in the range $[\alpha_1, \alpha_2]$, so we let $\gamma \leftarrow \gamma/2$. After a while $\gamma \rightarrow 0$.
>
>
> 4. *Minor remarks.* These were all helpful. We have opted to keep Figure 1 on the first page and instead clarified what outer steps are in the caption. We followed your advice on all other points.

---

> > ### Comment · AnonReviewer1 · 2020-11-20
> > **BOHB**
> >
> > Thanks for your reply.
> >
> > I'm not sure I follow this statement: "For instance, BOHB searches hyperparameters in narrow ranges that only contain reasonably good values while our experiment makes much fewer assumptions about the hyperparameters we are searching." BOHB doesn't assume that interval of alpha, what are the authors referring to? I understand that some previous work used those range of values but that means that the authors of that work were using a prior over the search space.
> >
> > I'm also very surprised that the authors say that BOHB doesn't match RS. There is quite some evidence in the literature that BOHB beats RS in a variety of applications. The BOTorch framework that the authors used seems not to be the best choice for the HPO of DNNs, BOHB would be a better choice.
> >
> > But anyway, a better baseline than RS would be ASHA or its predecessor Hyperband. These are based on RS but use successive halving to improve statistical efficiency.

---

> > > ### Author Response · Authors · 2020-11-23
> > > **Re: BOHB**
> > >
> > > Thank you for following through. We've had time to run BOHB experiments, and hope to expend on our last comment below.
> > >
> > > The range of alpha stated previously was taken from the tables in the appendices of the BOHB paper (Table 1,3,4). Note that the range for the CIFAR-10 experiment doesn't seem to be given, so we stated the range used for alpha in related experiments in the paper.
> > >
> > > Our last comment was based on BO experiments only, and assuming BOHB would suffer from similar issues. Since then, we ran BOHB and Hyperband experiments using the same search range, network and computational budget as our method. We use the HpBandSter library (https://automl.github.io/HpBandSter/build/html/index.html) which provides both methods. We use all standard BOHB hyperparameters and set  min budget equal to $5$ epochs.
> > >
> > >  On CIFAR-10, we obtained $\sim 87$% with BOHB and $\sim 85$% with Hyperband. Although more seeds should be run, evidently you were correct in pointing out that BOHB is better than BO and RS (while still falling short compared to using gradients, since our method gets $\sim 89$%). These preliminary results also indicate that the 'HB' component of 'BOHB' is mostly responsible for the good performance in our setting. Indeed, setting min budget = max budget for BOHB recovers the poor performance we observed in BO.
> > >
> > >  Note that some for of successive halving/HB could be combined to our method in future work to make our approach cheaper. We also expect that the gap between our method and BOHB would broaden if more hyperparameters were searched (and shrink for less hyperparameters), since BO/RS based-methods typically struggle with many hyperparameters.
> > >
> > > In short, after running more experiments we now agree with Reviewer 1 that adding these extra baselines do make the paper more complete, and we plan to do so once final results have all come in. Again, we thank Reviewer 1 for taking the time to follow through on this discussion.

---

### Official Review · AnonReviewer4 · 2020-10-26
**Review Non-greedy Gradient-based Hyperparameter Optimization Over Long Horizons**

**Rating:** 7
**Confidence:** 2

**Review:**

Review Non-greedy Gradient-based Hyperparameter Optimization Over Long Horizons
The paper presents an approach to hyper parameter optimisation over long horizons avoiding greedyness. This is achieved by calculating gradients wrt to the hyper parameters and using the sign of the gradients to indicate convergence. The paper is well written and structured and the suggested approach shows promising results. I only have two minor comments/ suggestions.

Introduction:
After first sentence: there is an empty space missing: ").This"

Experiments:
Could you compare your results based on some Bayesian Optimization based algorithm and compare to the computational cost? If not, why?

---

> ### Author Response · Authors · 2020-11-18
> **Reply to Reviewer 4**
>
> Thank you for your positive comments.
>
> The typo you spotted has been corrected, thanks.
>
> Your comment on Bayesian Optimization is relevant and was also pointed out by other reviewers, so we have provided more details in a separate comment on this page.

---

### Official Review · AnonReviewer3 · 2020-10-28
**The paper propose a new technique to compute hypergradients, relying on hyperparameter sharing, in order to prevent gradient degradation. In addition authors proposed a different stopping criterion for the bilevel optimization problem, which is paramount in practice. The paper seems very very interesting from a practical point of view.**

**Rating:** 5
**Confidence:** 2

**Review:**

Overall I vote for (weak) rejecting. IMO the main weakness of the paper is the clarity, for instance I had to read the article multiple times before understanding the main contribution. Some crucial terms should may have to be mathematically defined in the paper, like 'greedyness' for hyperparameter optimization, a term a was not aware of.


Advantages of the paper:
- the proposed algorithm seems to lead to significance gain in performance in practice
- authors proposed a new stopping criterion which seems to be more efficient in practice


Concerns:

1- The notion of greedyness is paramount for the paper, however it is not defined. Moreover the references provided are not very helpful to understand the concept. The word 'greedy' appears once in [1], twice in [4] and is not defined properly. The word 'greedy' does not appear in [2, 3]. Would it be possible for the authors to define the concept properly in the paper, or the provide a selfcontained reference?

2- I founded the main contribution of the paper hard to find. If I understood well, authors propose a new to estimate the hypergradient by averaging out, and then updating the hyperparameter. This is very subjective, I would recommend to highlight this contribution, and maybe to remove 'we combined the above [...] with momentum decay' in the introduction.

3- The experiments do not seem clear to me, I do not know if it comes from my lack of experience in the field of from the lack of clarity of the paper, but experiments were hard to follow. In particular the number of steps in the inner problem is provided, but what is the size of the hyperparameter searching space? In particular all the experiments are provided with convergence as a function of the number of steps. Since authors rely on forward differentiation, one step of the proposed algorithm can be more costly than one step of the algorithms in the baseline. Am I missing something? Is forward differentiation paramount for the proposed algorithm?

4- Authors claim that hyperparameter sharing is equivalent to averaging (page 5). Is it mathematically grounded? Is it trivial? Or could authors provide a reference?

There are a lot of things I did not understand, thus I voted for weak reject. However if authors answer my questions I am of course willing to change my score.


Minor:
- in section 1 there should have a space after the point in l3


[1] Luketina, J., Berglund, M., Greff, K., & Raiko, T. (2016). Scalable gradient-based tuning of continuous regularization hyperparameters. ICML 2016

[2] Franceschi, L., Donini, M., Frasconi, P., & Pontil, M. (2017). Forward and reverse gradient-based hyperparameter optimization. ICML 2017

[3] Baydin, A. G., Pearlmutter, B. A., Radul, A. A., & Siskind, J. M. (2017). Automatic differentiation in machine learning: a survey. JMLR

[4] Donini, M., Franceschi, L., Pontil, M., Majumder, O., & Frasconi, P. (2019). Scheduling the Learning Rate via Hypergradients: New Insights and a New Algorithm. arXiv preprint

---

> ### Author Response · Authors · 2020-11-18
> **Reply to Reviewer 3**
>
> We thank you for your feedback, and taking the time to read our submission several times. We hope to clarify some elements of our methods in what follows.
>
>
> 1. We use the term "greedy" the same way it is sometimes used in computer science, to describe "any algorithm that follows the problem-solving heuristic of making the locally optimal choice at each stage" (see Wikipedia page for Greedy Algorithm). Consider training a network on CIFAR for 50 epochs. Now consider the process of learning the best learning rate to use for epoch 1. We can find this learning rate such that it minimizes validation loss after epoch 1, but this is greedy (local). Indeed, what we we really want to find is the learning rate for epoch 1 that minimizes the validation loss after 50 epochs (non-greedy). We briefly defined greediness as locality in the introduction, and formalized it at the end of section 3.1. We added some explanations in Related Work. Please let us know if we could further clarify things in some way.
>
>
> 2. You are correct that showing the benefits of averaging out hypergradients from contiguous hyperparameters is the main contribution of our paper, as stated in (1) at the bottom of our introduction. However, this only becomes relevant for long horizons (where gradient degradation occurs), for which reverse-mode differentiation cannot be applied. Therefore, integrating this as part of a forward mode algorithm in a way that works also made up other contributions of this paper
>
>
> 3. The size of the hyperparameter search space is given in Appendix B for all experiments. Since we share hyperparameters across steps, we can learn schedules over many steps by only learning a few values (5+1+1 in Fig1). You are correct when saying that one outer step can be more expensive for our method than other methods, but results in Table 1 are computed over the same GPU budget for each method (so more outer steps are taken for random search, since it's very cheap). Forward mode differentiation is indeed paramount to our algorithm, because reverse mode does not scale well in memory for long horizons (see beginning of section 3.2).
>
>
> 4. This is related to weight sharing. When a parameter is shared across two computational graphs (i.e. two operations), its total gradient is the sum of the gradients coming from each graph. Therefore sharing two hyperparameter is the same as summing the hypergradients they would each get individually if there was no sharing. The average can be obtained manually from the sum trivially. This property was checked using the Pytorch autodiff.
>
>
> We hope that this helps clarify things. If this wasn't enough to address your queries, please let us know the points that remain unclear and we would be happy to pursue this discussion further. Again, we thank you for time you invested in reading our paper.

---

### Official Review · AnonReviewer2 · 2020-10-28
**Reviewer 2**

**Rating:** 6
**Confidence:** 4

**Review:**

The paper proposes an algorithm for tuning hyperparameters over long optimization problems to mitigate sub-optimal greedy solutions from short-horizon bias.  This is done by:

(1) sharing hyperparameters across optimization steps which reduces the variance of hypergradients and computational cost

(2) A method for assessing when our hyperparameters have converged

(3) A forward-mode gradient calculation for SGD with momentum

These components are combined into an algorithm and empirical results are presented.



Strengths:

The paper is well-written

The contributions are well-placed with the related work

Sharing hyperparameters across multiple steps is well-motivated.

The experiments on hypergradient variance are a useful diagnostic for these kinds of algorithms.

Tuning hyperparameters like this could be impactful if we scale it to state-of-the-art models and optimization procedures/horizons.


Weaknesses:

The forward-mode algorithm is limited to SGD with momentum and weight-decay.  More complicated optimization algorithms are used in many domains.

It’s not clear to me the sign of the hypergradient is a good test for convergence.

If we can’t use modern optimizers like Adam with the hypergradient, I am worried that the algorithm is very brittle.

The improvements aren’t very large relative to the hand-tuned baseline.  See CIFAR-10 where the baseline is 89.2 +- .2 and theirs is 89.3 +- .1.  Perhaps your algorithm may find this setting faster since there are about 100 runs for the hand-tuning.  I would want to see if there is some way to offer an improvement over the grid-search.  What happens if you start your training with the hyperparameters initialized at the best-value from the grid-search?

Only a handful of hyperparameters are tuned -- perhaps this is due to how the gradient calculation scales with the number of hyperparameters.  In this hyperparameter regime, a good baseline to test against might be bayesian optimization as opposed to grid-search.


My recommendation for the paper is a 6 -- i.e., marginally above the acceptance threshold. This is because the paper makes useful contributions toward optimizing hyperparameters over long-horizons, by motivating sharing hyperparameters via gradient variance and providing a forward-gradient calculation skeleton.  However, the current improvements relative to the baseline are mediocre, and it’s not clear how to scale the method to optimizers besides SGD with momentum.


The following points may help improve the paper, but did not affect my score:

Figure 1 says “10, 50 and 50”.  Is there a typo here?

---

> ### Author Response · Authors · 2020-11-18
> **Reply to Reviewer 2**
>
> Thank you for reading our work carefully and providing well thought-out comments. We hope to address your points in what follows.
>
> 1. *The forward-mode algorithm is limited to SGD with momentum and weight-decay*. While we chose to only show results on the SGD optimizer with momentum and weight decay, our method is applicable to any differentiable hyperparameter, which includes most modern optimizers. We focused on the optimizer that is by far the most common for CIFAR-10 so that the reader has a feel for what sensible hyperparameter values are, which hyperparameters are useful etc. while looking at our results. Other optimizers have different matrices $A, B, C$ but their derivation is analogous to that shown in Appendix A. We have clarified this point at the end of section 3.2.
>
>
> 2. *It’s not clear to me the sign of the hypergradient is a good test for convergence. If we can’t use modern optimizers like Adam with the hypergradient, I am worried that the algorithm is very brittle*. The hypergradient signs was observed to make convergence faster than other common optimizers. Hypergradients don't have the same distribution as common neural network gradients, and as such common optimizers are not the most appropriate. For instance, the hypergradient on the first step can be much larger than subsequent steps, because the first step has zero learning rate and therefore a small positive change in learning rate creates a large change in the validation loss. As such, using a momentum-based outer optimizers like Adam can bias subsequent gradient steps, and slow down outer optimization. This is not the result of unstable hypergradients, but rather a result of their natural distribution.
>
>
> 3. *The improvements aren't very large relative to the hand-tuned baseline*. This is correct. For the hand tuned baseline we searched ``around'' the standard hyperparameter settings used in the literature for CIFAR-10 and Wide-ResNets. It is hard to state how much time the community has collectively invested to settle on these hyperparameters, but we considered them to be an approximate upper bound for this dataset/network. Therefore the argument here is that our method can match the performance of this upper bound while other search methods fail to. While our hand tuned baseline did take more time to run, we didn't make that argument since it all depends on the size and quality of the search range chosen, which could be made arbitrarily large probably without improving the performance of the setting found.
>
>
> 4. *A good baseline to test against might be Bayesian optimization as opposed to grid-search*. Other reviewers have made this point so we have included our reply as a separate comment on this page.
>
>
> 5. *What happens if you start your training with the hyperparameters initialized at the best-value from the grid-search?*. We have tried this setting and did not find a significant improvements when considering several seeds. This seems to come from the fact that the scale of the noise on the validation loss irons out the margin for improvement around the hand-tuned configuration. In other words, small improvements can be observed for a given train-val split, but no significant improvement is found that holds for any train-val split.
>
>
> 6. *Typo in Figure 1?*. This was not a typo but our phrasing was somewhat confusing and has been clarified. Thank you.

---

### Author Response · Authors · 2020-11-18
**Bayesian Optimization Baseline**

Several reviewers have rightly pointed out that we could include Bayesian optimization (BO) as a potentially stronger baseline than random search. We have, in fact, run numerous BO experiments for our experimental configuration. We used the highly efficient BoTorch-Ax library, with the common BO settings found in the HPO literature: Gaussian process with Matern 5/2 kernel, with expected improvement (EI) as the acquisition function, and using Sobol search for the first few arms.

In the paper, we sought to showcase the efficiency of using gradients in HPO by considering a large hyperparameter search range. This means that we assume very little about the hyperparameters we are trying to learn. For instance we learn the learning rate in a range $\alpha \in [-1, 1]$ while a common search range in BO would be about $20$ times smaller like $\alpha \in [10^{-6}, 10^{-1}] $ [1], and would use a log scaling prior to search this space in a way that is tailored for learning rates. In our setting, we find that BO and related methods struggle to even match random search. This is because the search space mostly contains configurations that have a high validation loss, and BO spends much of its time exploring irrelevant fluctuations in that high loss region. In order to slightly outperform random search with BO, we had to divide the search range by $\sim 20$ and use $5$ times the computational budget. Indeed BO is usually orders of magnitude more expensive to run than our method, with some methods reporting up to $33$ GPU days on CIFAR-10 [1] while our algorithm takes $\sim 13$ hours (albeit for smaller architectures).  Finally, note that random search has often been found to match or outperform more complex search algorithms in follow up work, pointing to the fact that the success of HPO algorithms can often be explained by a narrow and contrived search range [2] [3].

We have included this information in Appendix E and pointed to it from the main text. We thank the reviewer for making this point and thereby anticipating future reader comments. We are happy to pursue this discussion further if needed.

**References**

[1] Falkner, Stefan, Aaron Klein, and Frank Hutter. "BOHB: Robust and Efficient Hyperparameter Optimization at Scale." ICML. 2018.

[2] Liam Li and Ameet Talwalkar, "Random Search and Reproducibility for Neural Architecture Search", 2019.

[3] Kaicheng Yu and Christian Sciuto and Martin Jaggi and Claudiu Musat and Mathieu Salzmann, "Evaluating The Search Phase of Neural Architecture Search" ICLR 2020.

---

### Decision · Program_Chairs · 2021-01-07
**Final Decision**

**Decision:**

Reject

**Comment:**

This paper investigates methods for gradient-based tuning of optimization hyperparameters.  This is an interesting area, and the paper isn't bad.  The examination of hypervariance seems relatively novel and useful.  I also appreciate the point about Bayesopt sometimes working well simply due to small ranges.

However, I agree with the criticisms of the reviewers.  Overall this paper isn't quite clear, thorough and impactful enough to make it in this round, but I think with more attention to baselines and scope this paper could be acceptable.

Some minor comments:

1) The signed-based optimizer, while simple and sensible (which is good), seem kind of ad hoc.
2) The authors don't seem to have properly scoped the problem and method, since greediness is only a major concern for inner optimization hyperparameters specifically.  It's not clear that for regularization parameters that this problem exists or that your method would apply.

A small nit:  Is hypervariance the right thing to look at, since the problem can exist even in deterministic settings?  Perhaps some sort of sensitivity analysis would be more appropriate.  Also you should reference Barack Pearlmutter's thesis which first explores these issues.   I would also mention that the hypervariance is generally tiny for smaller-than-optimal learning rates, and massive for larger-than-optimal learning rates, (the chaotic regime).